# Efficient CRISPR-Cas9 based cytosine base editors for phytopathogenic bacteria

Chenhao Li[1,2,7], Longfei Wang[1,7], Leland J. Cseke[1], Fernanda Vasconcelos[1], Jose Carlos Huguet-Tapia[3], Walter Gassmann [1], Laurens Pauwels [4,5], Frank F. White [3], Hansong Dong[2] & Bing Yang [1,6✉]

Phytopathogenic bacteria play important roles in plant productivity, and developments in gene editing have potential for enhancing the genetic tools for the identification of critical genes in the pathogenesis process. CRISPR-based genome editing variants have been developed for a wide range of applications in eukaryotes and prokaryotes. However, the unique mechanisms of different hosts restrict the wide adaptation for specific applications. Here, CRISPR-dCas9 (dead Cas9) and nCas9 (Cas9 nickase) deaminase vectors were developed for a broad range of phytopathogenic bacteria. A gene for a dCas9 or nCas9, cytosine deaminase CDA1, and glycosylase inhibitor fusion protein (cytosine base editor, or CBE) was applied to base editing under the control of different promoters. Results showed that the RecA promoter led to nearly 100% modification of the target region. When residing on the broad host range plasmid pHM1, CBE$_{RecAp}$ is efficient in creating base edits in strains of *Xanthomonas*, *Pseudomonas*, *Erwinia* and *Agrobacterium*. CBE based on nCas9 extended the editing window and produced a significantly higher editing rate in *Pseudomonas*. Strains with nonsynonymous mutations in test genes displayed expected phenotypes. By multiplexing guide RNA genes, the vectors can modify up to four genes in a single round of editing. Whole-genome sequencing of base-edited isolates of *Xanthomonas oryzae* pv. *oryzae* revealed guide RNA-independent off-target mutations. Further modifications of the CBE, using a CDA1 variant (CBE$_{RecAp}$-A) reduced off-target effects, providing an improved editing tool for a broad group of phytopathogenic bacteria.

[1] Division of Plant Science and Technology, Bond Life Sciences Center, University of Missouri, Columbia, Missouri, USA. [2] Department of Plant Pathology, Nanjing Agricultural University, Nanjing, Jiangsu, P. R. China. [3] Department of Plant Pathology, University of Florida, Gainesville, Florida, USA. [4] Department of Plant Biotechnology and Bioinformatics, Ghent University, B-9052 Ghent, Belgium. [5] Center for Plant Systems Biology, VIB, B-9052 Ghent, Belgium. [6] Donald Danforth Plant Science Center, St. Louis, Missouri, USA. [7] These authors contributed equally: Chenhao Li, Longfei Wang. ✉email: yangbi@missouri.edu

Bacteria, including beneficial symbionts and pathogens, play essential roles in our life and the health of ecosystems[1–3]. When causing disease, plant-associated bacteria, the so-called phytopathogenic bacteria, can cause tremendous reduction in crop yield. Understanding the genetic basis of pathogenicity provides insight into host-pathogen interactions and to conceive strategies and management plans for the effective control of crop diseases[2]. Methods for targeted genetic analysis traditionally depend on homologous recombination process, which, in many cases, is time-consuming and inefficient[4]. Innovative emerging molecular biotechnology offer the promise of precise and speedy research tool for analyzing genes of interest in phytopathogenic bacteria.

CRISPR-Cas9 based genome editing has emerged as revolutionary genome engineering tools, applicable to both eukaryotes and prokaryotes[5–8]. Genome editing with CRISPR-Cas was initially based on the abilities of Cas proteins to introduce double-stranded DNA breaks (DSBs), which, in turn, trigger the host DSB repair by either non-homologous end joining (NHEJ), or homology directed repair (HDR)[9]. Ironically, innate CRISPR systems, while of bacterial origin, have yet to be widely applied for phytopathogenic bacteria. Bacteria lack the NHEJ pathway, and DSBs lead to cell death, limiting applications[10]. HDR depends on sister chromatids or exogenously provided homologous template DNA fragments, which limits broad applications[9]. The system can be used for selection of bacteria that have experienced loss of the targeted region based on the repair template through HDR[11].

DSB-free CRISPR-Cas9 based editing systems have been developed for genome editing in a variety of bacterial species[12–15]. Base editors enable the conversion of one targeted base to another without the need for DSB or utilizing a donor repair template. The base editors are composed of an enzymatically defective Cas protein, typically, dCas9 (dead Cas9) or nCas9 (Cas9 nickase), and a cytidine or adenosine deaminase, resulting in cytosine base editors (CBE) or adenine base editors (ABE), respectively. CBEs generate C to T substitutions by C-to-U deamination and subsequent DNA replication, while ABEs mediate the conversion of A to G by A-to-I deamination[16,17]. CBE has been more widely used in industrially and clinically relevant microorganisms than ABE, which is likely due to the higher editing efficiency and high GC content of bacterial genomes[18]. Use of base editors, specifically, in phytopathogenic bacteria has been limited to *Agrobacterium* and *Pseudomonas*[13,14]. Here, a CRISPR-Cas9 base-editing system was designed from a system developed for *Agrobacterium* for application in a broader range of phytopathogenic bacteria.

## Results

### Establishment of a *Xanthomonas* compatible base editing system.
A previous *Agrobacterium* CBE (dCas9-CDA1-UGI) was based on a binary vector to express the guide RNA and fusion protein of dCas9, the cytidine deaminase from *Petromyzon marinus* and uracil glycosylase inhibitor domain (dCas9-CDA1-UGI), and the *sacB* gene. Expression of *sacB* in the presence of elevated sucrose levels is lethal, possibly due to toxic levels of fructose polymers, ultimately, enabling selection for strains that have lost the plasmid in the population[13]. The destination vector pVS1, with the oriV origin for replication, has limited host range among bacteria[19], providing replication and stability in *Agrobacterium*, while not in *Xanthomonas*[20]. To modify the *Agrobacterium* CBE, the destination vector was replaced with the *Xanthomonas*-compatible vector pHM1, a broad host range plasmid derived from pR140 and containing pSa *ori*[21]. Guide RNA (gRNA) was expressed under the control of the synthetic promoter J23119 of the original vector, and a gRNA targeting the

*X. oryzae* pv. *oryzae* (*Xoo*) sucrose utilization gene *suxC* from the *Xoo* strain PXO99[A] was used as a test gene. Loss of *suxC* leads to loss of sucrose utilization and a distinct phenotype when the mutants are grown on sucrose as the sole carbon source. No loss of sucrose utilization was observed, and genotyping of transformants derived from pHM1-CBE-gSuxC did not reveal any mutants with a C to T transition. The dCas9-CDA1-UGI gene is under control of *virB* promoter (VirBp) of *Agrobacterium*, and VirBp might not provide adequate expression levels of dCas9-CDA1-UGI in *Xoo*. Six *Xanthomonas* gene promoters from PXO99[A] were individually used to replace the VirBp (Supplementary Fig. 1a). Ligation products with the promoter XopZp or PIP1p failed to produce *E. coli* transformants after multiple attempts, while cloning attempts with the promoters HrpXp, RecAp, PIP2p and PIP3p were successful, resulting in the four plasmids HrpXp-dCas9-CDA1-UGI, RecAp-dCas9-CDA1-UGI, PIP2p-dCas9-CDA1-UGI and PIP3p-dCas9-CDA1-UGI (Supplementary Fig. 1b).

Two gRNAs were designed that would lead to a stop codon in *suxC* and *suxB*, respectively, in their protospacers at 17–19 bp upstream of the PAM. The *suxC* encodes a sucrose transporter and *suxB* encodes an amylosucrase for the *Xanthomonas* sucrose utilization, and the inactivation of either one results in a phenotype of small colonies on sucrose medium[22]. Each of the two gRNAs (gSuxC and gSuxB) in combination with each of the four different promoters driving dCas9-CDA1-UGI were tested by introduction into PXO99[A] and growth on nutrient broth medium without sucrose (Fig. 1a, b).

Three transformants from each construct were selected for sequence analysis of PCR-amplicons from the targeted regions of *suxC* and *suxB* separately. The percentages of peaks for the target nucleotides in the sequencing chromatograms were evaluated for base composition and percentage of C to T. The construct with RecAp showed the highest editing efficiency with 100%, 100% and 55.4%, respectively, for C to T transition of the three C's, C20, C18, and C14, relative to the PAM site in *suxC* (Supplementary Fig. 2a, c). The HrpXp-based construct produced 80.0%, 62.2% and 21.5%, respectively, C to T at the same positions. Both PIP2p- and PIP3p-based CBEs yielded 100% editing efficiency for C's at C20 and C18, while at 18.9% for PIP2p and 45.7% for PIP3p at C14 (Supplementary Fig. 2a, c). The C to T transition at position 14 led to a premature stop codon in *suxC* (Supplementary Fig. 3a). Constructs with gSuxB resulted in conversion frequencies slightly lower than those for gSuxC (Supplementary Fig. 2b, d). The C to T change at C16 led to a premature stop codon in *suxB* (Supplementary Fig. 3b).

One *Alwn* I restriction site is present in the sequence spanning C10 to C16 in the target site of *suxB*. *Alwn* I digestion of the PCR products was also used to assess the C to T transition frequency at C16 in *suxB*. Twelve selected isolates derived from individual gSuxB constructs showed higher base editing efficiency with RecAp based on the loss of the *Alwn* I site (Supplementary Fig. 4a). After evicting the CBE plasmids based on *sacB*-mediated selection, all isolates contained the edited stop codons at the desired target sites in *suxC* and *suxB*, showing a defective growth phenotype in sucrose medium (Supplementary Fig. 4b). Thus, the RecAp-based CBE, referred to as CBE$_{RecAp}$, was selected for further experiments. NBLAST sequence searches of the NCBI database indicated that the RecAp was restricted to *Xanthomonas* species at the DNA level, despite RecA itself being a conserved factor in homologous recombination and DNA-damage repair in all bacteria (Supplementary Fig. 5).

To quantify the editing efficiency in *Xoo* by the RecAp-based CBE more accurately, the constructs CBE$_{RecAp}$-gSuxB and CBE$_{RecAp}$-gSuxC were introduced into PXO99[A], fifteen single colonies (three pools with 5 colonies each) at day 4 after

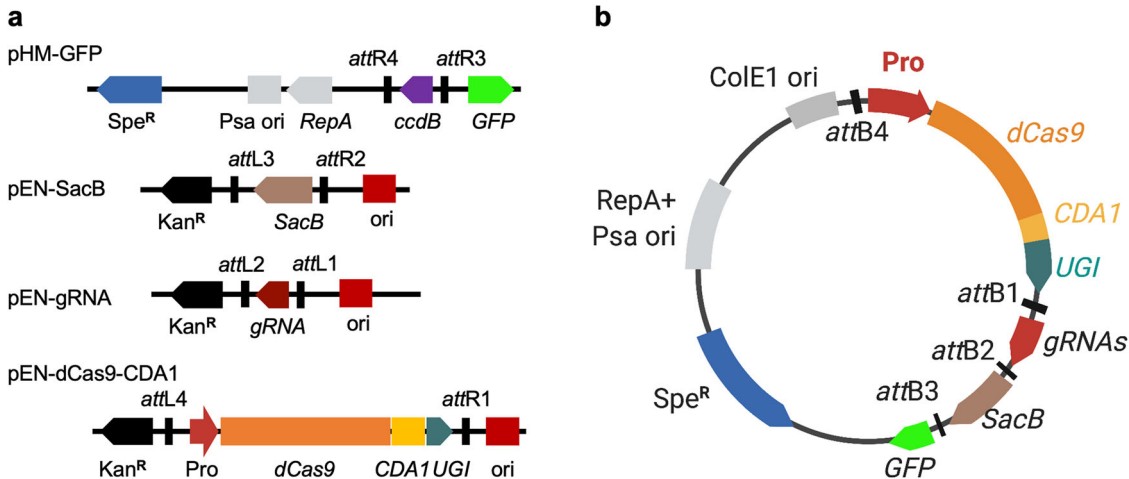

**Fig. 1 Phytopathogenic bacterial cytidine base editor system. a** A four-module system to make gene specific cytidine base editor. Expression cassettes of guide RNA (gRNA), *SacB*, and chimeric gene of dead Cas9 (dCas9), cytidine deaminase (CDA1) and DNA uracil glycosylase inhibitor (UGI) each are flanked by the Gateway recombination motifs (*att*R and *att*L). **b** Schematic map of single construct expressing dead Cas9 deaminase, guide RNA expression cassette and GFP. Pro, promoter derived from *Xanthomonas*. dCas9, a nuclease dead Cas9; CDA1 *P. marinus* cytidine deaminase, UGI uracil DNA glycosylase inhibitor, gRNA guide RNAs, *SacB* the counter-selectable marker for plasmid curing after editing and GFP driven by *E. coli* glT promoter. The plasmid contains the spectinomycin-resistant (Spe[R]) gene, the pSa origin and RepA, high-copy number origin of ColE1.

transformation were subjected to PCR-amplification of relevant regions and deep sequencing using MiSeq. The editing frequencies for C18 and C16 in *suxB* reached more than 98.2% and 86.1%, respectively (Fig. 2a), while frequencies for C20, C18 and C14 in *suxC* were 99.9%, 99.1% and 68.8%, respectively (Fig. 2b). Therefore, the data were consistent in editing efficiencies based on both Sanger sequencing and deep sequencing of multiple clones.

To facilitate identification of the initial transformants and subsequently CBE-evicted isolates, a green fluorescence protein (GFP) gene was incorporated in pHM1. The *E. coli* glpT (sn-glycerol 3-phosphate transport) gene promoter and the super-folder GFP gene[23] was subcloned into the backbone of the destination vector pHMattR3-attR4. When introduced into *Agrobacterium* strain LBA4404, the green fluorescence could be easily detected by epifluorescence (Supplementary Fig. 6a, b). After counterselection with 10% sucrose, single colonies without fluorescence were identified and further confirmed to lack CBE (Supplementary Fig. 6c, d).

**Efficient base editing in *Pseudomonas* and *Erwinia*.** The CBE$_{RecAp}$ was tested in *Pseudomonas syringae*. Two type-III effector genes, *avrPto* and *hopK1* in *P. syringae* pv. *tomato* DC3000, were selected as the target due to known phenotypes upon loss[24,25]. Use of the gRNA gAvrPto-1 led to 100% of C to T transition at C17 of the *avrPto* target site, while lower frequencies of conversion were observed at C20 and C14 as revealed by the deep sequencing data from 15 individual colonies at day 4 after transformation (Fig. 2c, d). Similarly, a second gRNA for *avrPto* (gAvrPto-2) yielded base edits at C20, C16 and C13 at 72.5%, 99.8% and 53.2%, respectively (Supplementary Fig. 7a, b). After eviction of CBE plasmids, two types of DC3000 mutants, one encoding only 42 aa and another encoding 85 aa, were obtained, reflecting gene truncations at C17 and C16, respectively (Fig. 3a). Bacterial growth populations and disease assays were performed in Arabidopsis accession Bu-22[26]. Both mutant strains lost the abilities to trigger AvrPto-dependent resistance in Bu-22 and showed enhanced bacterial populations and associated chlorosis of inoculated leaves (Fig. 3b–d).

Two gRNAs targeted *hopK1*, whose presence has a phenotypic effect in Arabidopsis accession Col-1[27]. The two gRNAs (gHopK-1

and gHopK-2) were introduced into DC3000, and deep-sequencing results of PCR-amplicons from 15 clones showed 100% editing rate at C16 and C19, and more than 68% at C20 by gHopK-1, while gHopK-2 produced 100% editing frequency at the targeted C18 (Supplementary Fig. 7c, d). The gRNA gHopK-1 was expected to induce two premature stop codons (corresponding to Q61 and R62), and gHopK-2 was predicted to induce one stop codon (corresponding to Q155) (Supplementary Fig. 7e, f). The mutant isolates, after CBE plasmid eviction, were genotyped, and bacterial populations and disease symptom assays were determined after inoculation into Arabidopsis Col-1. The *hopK1* mutants from either gRNA displayed decreased bacterial population and reduced disease symptoms (Supplementary Fig. 8a–c).

CBE$_{RecAp}$ was then tested in *Erwinia amylovora*, the causal agent of fire blight in apples and pears[28,29]. The gene for the type-III effector gene *dspA/E* in *E. amylovora* strain Ea9 was targeted due to the requirement of the gene for the ability of Ea9 to trigger cell death in nonhost tobacco[28]. Two gRNAs were constructed corresponding to *dspA/E*. The gDspA/E-1 was designed to convert either the codon CGA (Q239) or CAA (R240) to a stop codon (Fig. 2e, f), while gDspA/E-2 targeted the antisense strand, changing the codon TGG for W433 to a stop codon efficiently (Supplementary Fig. 9a, b). Other C residues within the protospacers were also edited, indicating CBE$_{RecAp}$ was operating efficiently in *Erwinia* as demonstrated by deep sequencing of PCR-amplicons from 15 clones (Fig. 2e; Supplementary Fig. 9a). Two types of mutants encoding the expected 238 aa and 432 aa truncated protein products of the wild type 1833 aa protein were obtained, respectively, due to two gRNAs (Fig. 4a). The wild-type Ea9 triggered a HR, while the two mutants failed to do so in *N. benthamiana* 24 h post-inoculation (Fig. 4b).

As previous studies demonstrated, Cas9 nickase (nCas9) could promote increasing of base editing efficiency than dead Cas9 in human cells[16], nCas9-containing CBE$_{RecAp}$ was developed by restoring histidine at position 840 of dCas9. nCas9 containing CBE was tested in *Pseudomonas*, *Xanthomonas* and *Erwinia*, but no significant difference was observed in the optimal editing window, namely C16 to C19, compared to dCas9-containing CBE (Supplementary Fig. 10 for *Pseudomonas*; Supplementary Fig. 11 for *Xanthomonas* and *Erwinia*). However, the editing window has been extended by nCas9, as shown by gHopK-2 in *Pseudomonas*. nCas9-

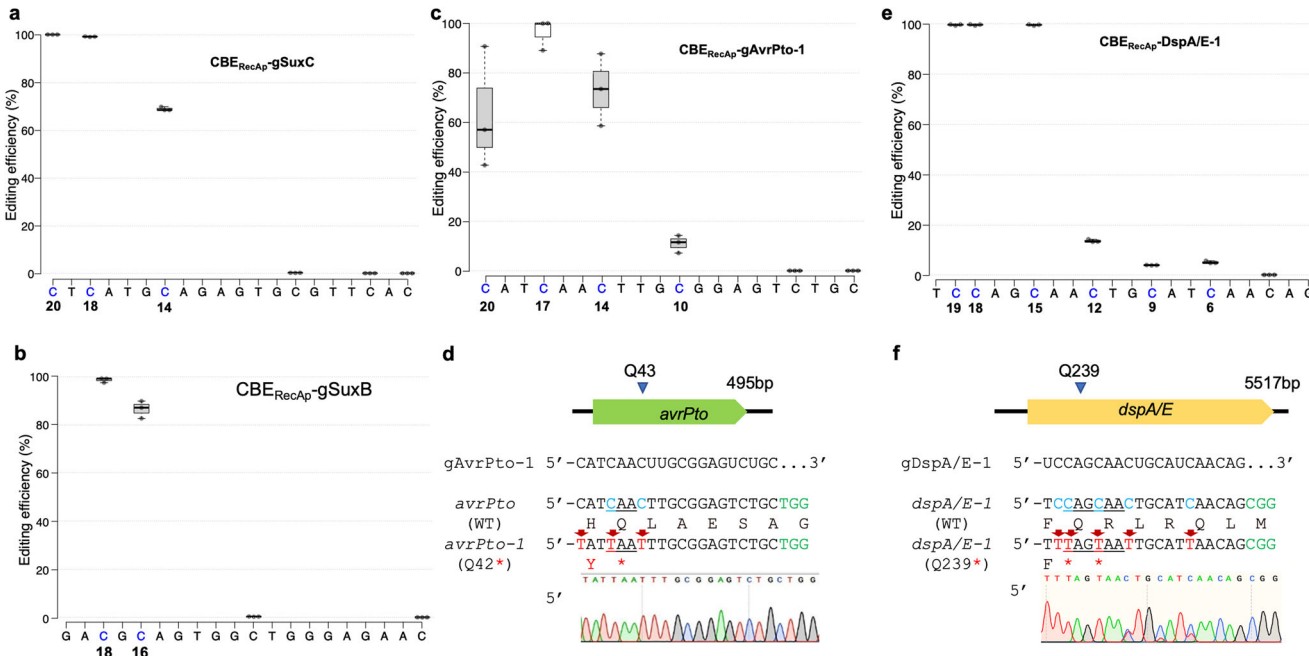

**Fig. 2 Efficient base editing in *Xanthomonas*, *Pseudomonas* and *Erwinia*. a** Quantification of the C to T editing before curing for gSuxC in PXO99[A]. dCas9-CDA1-UGI was driven by promoter RecAp. The percentage of C to T conversion is based on the deep sequencing analysis. Edited C's in the protospacer are indicated by blue color with the numbers for the positions relative to PAM. **b** Quantification of the C to T editing before curing for gSuxB in PXO99[A]. The percentage of C to T conversion is based on the deep sequencing analysis of PCR-amplicons of target regions. **c** Quantification of the C to T editing before curing for gAvrPto-1 in *Pseudomonas* DC3000. dCas9-CDA1-UGI was under control of promoter *RecA*. **d** Base editing outcomes after eviction of editor. **e** Quantification of the C to T editing before curing for gDspA/E-1 in *Erwinia* Ea9. dCas9-CDA1-UGI was under control of promoter *RecA*. **f** Base editing outcomes after eviction of editor.

based CBE produced a significantly higher editing rate at C10, C12 and C13 than dCas9-containing CBE (Supplementary Fig. 10).

**Multiplexing base editing in *Pseudomonas* and *Xanthomonas*.** To broaden the application of CBE$_{RecAp}$, five individual gRNA shuttle plasmids were designed to construct multiplexing gRNAs in CBE$_{RecAp}$. The plasmids pTL-Bag1 and pTL-Bag2T were designed for combining two gRNAs into a single CBE$_{RecAp}$ vector, while pTL-Bag1, pTL-Bag2, pTL-Bag3, and pTL-Bag4 were designed for combining four gRNAs into an individual vector, linking the modules together through the Golden Gate Assembly method[30] (Supplementary Fig. 12). The modular gRNAs each are driven by the same promoter J23119 and can be ligated together and into the recipient vector (pEN-J23119-sfGFP) in an orderly orientation through compatible 4-nt overhangs generated by *Bsa* I digestions (Supplementary Fig. 12; Supplementary Table 2).

The two-gRNA system was applied to the two *Xanthomonas* type III effector genes, *xopF* and *xopN* in PXO99[A]. The construct exhibited editing efficiency of 100% at the targeted codon at C17, converting the codon to a stop codon of both genes (Supplementary Fig. 13a–d). When applied in DC3000 and targeting *avrPto* and *hopK1* simultaneously with gAvrPto-1 and gHopK1-1, six selected isolates contained the desired mutations (Supplementary Fig. 14a, b).

Guide RNAs for four different type III effector genes (*avrBs2/xopR/xopP/xopZ*) were constructed into pTL-Bag1, pTL-Bag2, pTL-Bag3, and pTL-Bag4. Quadruple knockout isolates were obtained from PXO99[A] (Supplementary Figs. 15a–d; 16a–d). After plasmid eviction, another eight type III effector genes were altered in two additional rounds of editing (4 genes per round) (Supplementary Figs. 17a–d; 18a–d). Mutant isolates from each round were genotyped and assayed for virulence in rice, and

bacterial growth in the culture medium was measured. The mutant isolates grew similarly to the wild type strain in culture (Supplementary Fig. 19), while in the disease assay, the quadruple, octuple and duodecuple mutants displayed reduced virulence. The twelve-gene mutants had the greatest loss of virulence based on lesion length assay when compared to the parental PXO99[A] and progenitor mutant quadruple and octuple strains (Fig. 5a, b, c).

**Assessment of off-target mutations in edited bacteria.** Genome-wide off-target edits were examined in four edited strains of PXO99[A], including one clone of a *suxB* mutant, one clone of a *suxC* mutant, and two clones of a quadruple type III effecter gene mutant (*avrBs2/xopR/xopP/xopZ*). Deamination causes genome-wide mutations in a gRNA independent manner[31,32]. All off-target mutations are assumed to be gRNA-independent as no close gRNA sequences are present in the genome. Twenty-one and forty-two SNVs (single nucleotide variations) were detected in *suxB* and *suxC* mutants, respectively, in comparison to PXO99[A] (Supplementary Fig. 20a, b). All the SNVs represented C to T transitions. The sequences of the two quadruple mutant strains showed the selected clones had identical sequences and were likely siblings. When compared to the reference genome sequence of PXO99[A], 188 off-target sites were found in the quadruple mutant (Supplementary Fig. 20c). The off-target sites for all strains showed no preference for sites of mutation. Non-synonymous (amino acid changing) mutations occurred in about twenty-five percent of the SNVs (Supplementary Fig. 20d). Approximately half of the changes had the TC context, and the lowest off-target sites had the AC context (Supplementary Fig. 20e).

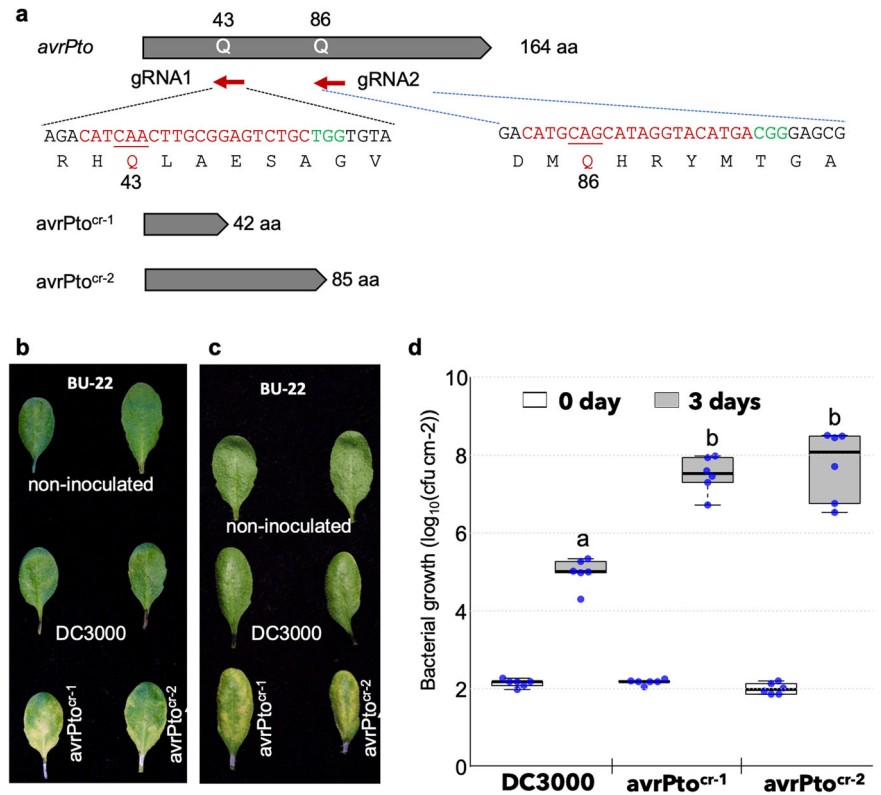

**Fig. 3 Base editing of *avrPto* in *Pst* DC3000. a** Schematic of *avrPto* and the CRISPR deletions caused by guide RNAs, gRNA1 and gRNA2, resulting in *avrPto*^cr-1^ and *avrPto*^cr-2^. **b, c** Replicated disease responses of Arabidopsis accession BU-22 when infiltrated at a bacterial density of $1 \times 10^6$ cfu/ml with DC3000 compared to decreased responses observed using *avrPto*^cr-1^ and *avrPto*^cr-2^. **d** In planta bacterial growth assay in BU-22 infiltrated with DC3000, *avrPto*^cr-1^ or *avrPto*^cr-2^ at a bacterial density of $5 \times 10^4$ cfu/ml. Values represent averages from two independent experiments with triplicate samples. Boxes extend from 25th to 75th percentiles and display median values as center lines. Whiskers plot minimum and maximum values and individual data points. Three independent experiments each with triplicate samples were performed, and treatments with different lowercases are significantly different at $p < 0.05$).

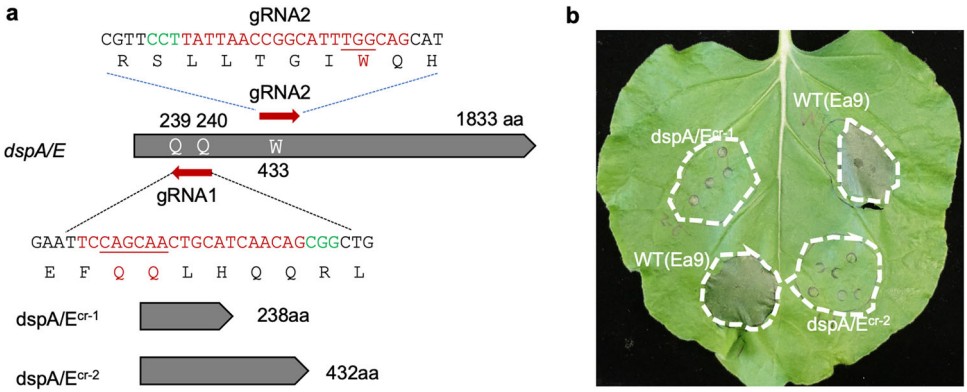

**Fig. 4 Erwinia *dspE/A* mutants lost the ability to induce HR in *N. benthamiana*. a** Schematic of *dspA/E* and the CBE-induced deletions resulting in *dspA/E*^cr-1^ and *dspA/E*^cr-2^. Q, glutamine; W, tryptophan. The numbers indicate the positions of triplets in the *dspA/E* coding sequences (in red). Two types of truncated mutants are shown. **b** *N. benthamiana* leaf 48 h after infiltration with parallel dilution series of suspensions of strains Ea9 (WT) and Ea9 *dspE/A* mutant 1 (239 aa) and mutant 2 (432 aa). The concentrations infiltrated are $5 \times 10^7$ cfu/ml.

**Engineering of CDA1 variant for reduced off-target editing.** Engineered APOBEC1 deaminases have variations that reduce deaminase off-target activity[33,34]. Based on the structural similarities to the rat deaminase APOBEC1 variants, CBE_RecAp system was modified by swapping the CDA1 portion with four CDA1 variants, including S30A (CDA1-A), S30A + H31A (CDA1-AA), W94Y + R133E (CDA1-YE), and W94Y + R133E + W139E(CDA1-YEE). Thymidylate synthetase (ThyA) is a highly conserved enzyme from bacteria to human, and catalyzes the conversion of deoxyuridine monophosphate (dUMP) to deoxythymidine monophosphate (dTMP)[35]. The *thyA*-deficient *Agrobacterium* cannot grow on drop-out media, without a supplement of thymidine. Thus, the survival rate of *Agrobacterium* carrying the base editor was used to determine the on-target editing activity of the various deaminases. One gRNA was designed to target *thyA*, which, with unmodified CBE_RecAp, induced efficient base editing in *Agrobacterium* LBA4404 at C17 and C16 to cause a premature stop codon (Supplementary Fig. 21). LBA4404 transformed with

 5

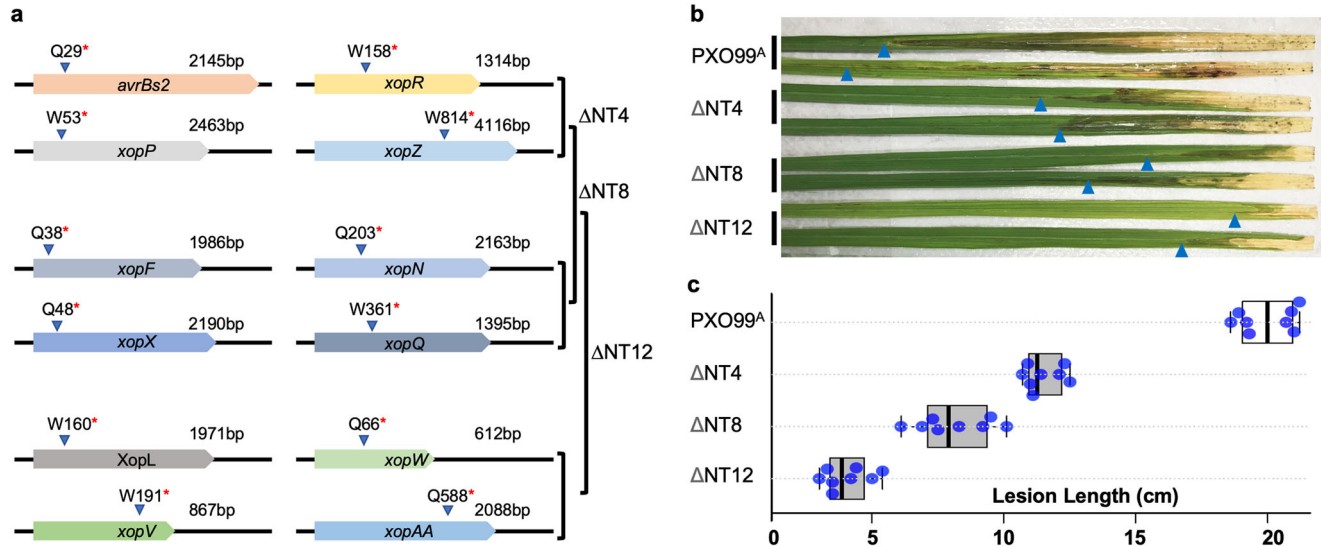

**Fig. 5 Type III effector gene mutants created by multiplex base editing exhibit reduced virulence in rice. a** Schematic structures (not in scale) of twelve type III effector genes with truncations as indicated by an asterisk for a premature stop codon. The quadruple (ΔNT4), octuple (ΔNT8), and duodecuple (ΔNT12) mutants were sequentially generated in three rounds. **b** Lesion phenotypes of Kitaake caused by the respective *Xoo* strains as indicated at the left side of leaves with arrow heads pointing to the edges of lesions. **c** Lesion lengths in leaves of Kitaake rice caused by the respective *Xoo* strains as indicated at the left side of graphs. Boxes extend from 25th to 75th percentiles and display median values as center lines. Whiskers plot minimum and maximum values and individual data points. *n* = ~20 sample points. Two biological replicates were performed with similar results.

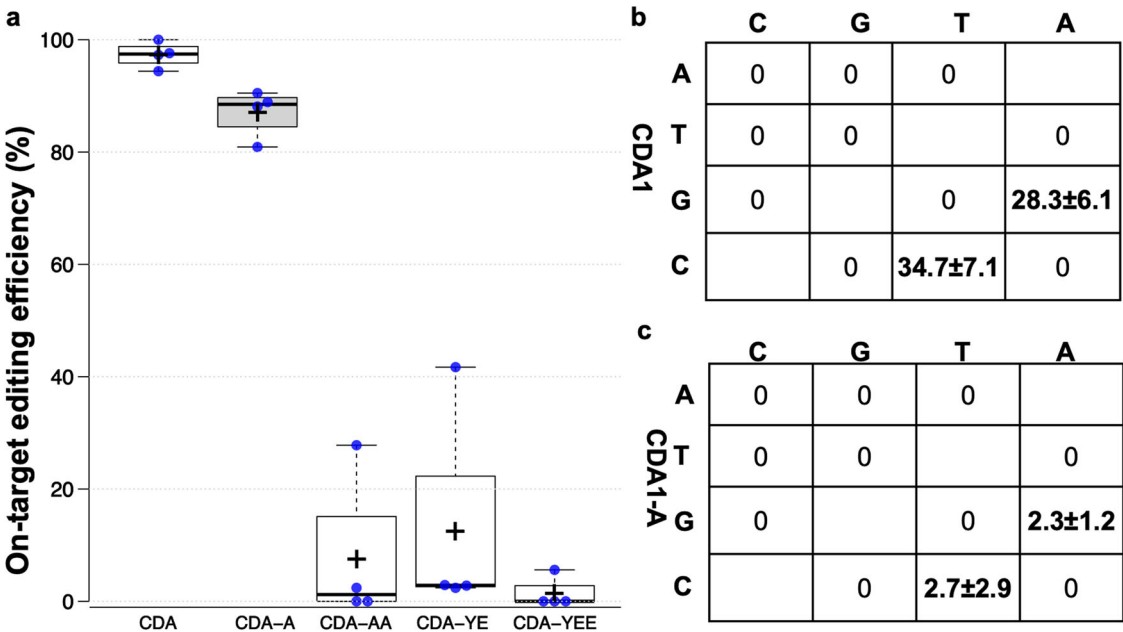

**Fig. 6 Assessment of the deaminase variants on- and off-target editing. a** On-target editing efficiency induced by CDA1 and its four engineered variants as indicated below boxes. Boxes extend from 25th to 75th percentiles and display median values as center lines. Whiskers plot minimum and maximum values and individual data points. Four biological replicates were analyzed for individual base editors. Tukey's Honest Significant Difference test was employed to determine the statistically significant differences between base editors. **b, c** Off-target C to T conversion caused by CDA1 (**b**) and its engineered variant (CDA1-A) (**c**) in *Agrobacterium* LBA4404. Three biological replicates were analyzed, and data represent the means ± standard deviations.

wild-type $CBE_{RecAp}$ has the lowest survival rate on thymidine drop-out media compared to the other deaminase variants (Supplementary Fig. 22). Amino acid changes in CDA1-AA, CDA1-YE, and CDA1-YEE compromised deaminase target activity, lowering the editing efficiency to less than 10% (Fig. 6a). In contrast, $CBE_{RecAp}$ with CDA1-A could maintain the on-target editing efficiency of 85% compared to the wild type CDA1 (Fig. 6a).

Consistently, the deep sequencing results revealed that $CBE_{RecAp}$ with CDA1-A remained efficient editing at C17, while the editing efficiency at C16 decreased to 43% compared to the wild type CDA1 (Supplementary Fig. 23a, b).

To assess $CBE_{RecAp}$-A off-target activity, three clones that were derived from either $CBE_{RecAp}$ or $CBE_{RecAp}$-A were whole genome sequenced and analyzed after eviction of the CBE plasmids. All

the off-target events from either treatment were C to T transitions. $CBE_{RecAp}$-A induced significantly lower SNVs than $CBE_{RecAp}$ (Fig. 6b, c; Supplementary Fig. 24). We next used $CBE_{RecAp}$-A to perform multiplex base editing of four *Xanthomonas* outer protein (xop) genes in *X. campestris* pv. *campestris*. The four Type III effector genes (*xopK, xopZ, xopA* and *xopL*) were successfully edited in a single clone, resulting in a quadruple knockout strain (Supplementary Fig. 25).

## Discussion

Here, we developed a versatile CRISPR-Cas9-based cytosine base editing system with high efficiency in various bacterial pathogens, including *Xanthomonas*, *Pseudomonas*, *Erwinia*, and *Agrobacterium*. Use of the *recA* gene promoter from *Xanthomonas*, combined with the broad- range vector pHM1, improved the editing efficiency and broadened the range of bacterial species. Multiplex base editing, here, with up to four sites, was also incorporated and demonstrated to work with high efficiency. The $CBE_{RecAp}$ vector should be applicable to other phytopathogens, although only tested here in four phytopathogenic bacterial species. The number of gRNA for multiplex genome editing may also be increased in the near future. The system has many advantages beyond gene knockouts, including the generation of marker-less mutations, and non-polar mutations in polycistronic operons. The near 100% efficiency may allow large scale or whole genome mutation approaches, targeting hundreds of genes and other DNA regions simultaneously. The vector also includes the *sacB* plasmid eviction component and GFP fluorescence-based identification of transformants and plasmid-cured edited cells.

A few superior features have been developed in our system, including a broad range vector, an improved promoter for dCas9-deaminase, multiplex guide RNA cloning vectors, green fluorescence-based identification of transformants and plasmid-cured edited cells, and lastly, the off-target improved deaminase. With our system, single and multiple (up to four) guide RNAs can be built into an intermediate guide RNA system. Gateway recombination enables 4 components (i.e., deaminase, guide RNA, *sacB* and *GFP* genes, and the broad range vector) to combine into a single plasmid before being introduced into bacteria of interest. GFP positive transformants are genotyped for site-specific base editing before using counter-selection of bacterial cells with sucrose to obtain plasmid free base edited clones that are sucrose tolerant and GFP negative. The feasibility and efficacy of our system, referred to as Phytobacterial Cytosine Base Editor (PCBE), have been demonstrated in *Xanthomonas oryzae* pv. *oryzae*, *Pseudomonas syringae* pv. *tomato*, *Erwinia amylovora*, and *Agrobacterium tumefaciens* with high-efficiency single and multiple gene editing. Comparative analysis of whole genome sequences of the edited and parental wild-type strains of *Xoo* and *Agrobacterium* revealed dramatic improvement in guide RNA-independent off-target effects in our system. One concern for base editing is the nonspecific activity of the deaminase, which can interfere with interpretations of phenotype. Whole genome sequencing indicates that off target mutations did not occur preferentially at specific cytosine residues and testing of multiple independent edited clones will ensure phenotypes are not due to off target mutations. Alternatively, gRNA-independent deaminase activity is not restricted to bacterial application and studies in other applications have provided that off target deamination can be reduced by engineered variants. Off target editing by $CBE_{RecAp}$ was reduced by 92% in *Agrobacterium* by incorporation of the CDA1-A substitution previously identified at the deaminase APOBEC1 to produce $CBE_{RecAp}$-A. Additional improvements are possible in the future. The UGI module could be modified to inhibit base excision repair for improvement of high fidelity; and

further engineering of deaminases could narrow the editing window to reduce collateral alteration to unintended target nucleotides[36–39]. The system could also be designed with dual base editors[40–42]. Finally, the location of editing is constrained by the availability of the PAM sequence to design guide RNA. The use of dCas9 variants for PAM-less and relaxed PAM sequence would be the choice for the base editor to circumvent this limitation[43–45].

Current base editors largely catalyze conversions (purine to purine and pyrimidine to pyrimidine); however, there are reported technologies for base transversions (purine to pyrimidine or pyrimidine to purine)[46,47]. The strategy used in base transversion mostly depends on the unstable DNA repair process. But the underlying mechanisms still are based on the error prone repair. Advancing the knowledge of DNA repair mechanisms will help to improve the development of base transversion. Primer editing is an advanced genome-editing technology that uses a prime-editing guide RNA as a reverse-transcription template[48]. The editing inefficiency is the primary restriction to wide application of prime editing in bacteria. However, prime editing is versatile in substituting insertion, deletion, and combinatorial editing without DSBs. The prime editing system is already reported in *E. coli*, but at low efficiency, especially for multiplex editing[49]. The expanded tools developed for microbial organisms, including phytopathogenic bacteria, will enhance the capacity of research on microbiology and gene editing biotechnology.

Application of multiplex base editing is particularly apropos for genetic analysis of plant pathogenic bacterial type III effector genes. Strains typically contain multiple genes, and the evidence indicates many function redundantly. With high-efficiency editing and high-throughput analysis, whole-genome metabolic reprogramming and protein evolution could be achieved in situ[50,51]. Taking the advantage of sgRNA library synthesis, gene evolution could be achieved in single bacterial cells[52].

## Methods

**Strains, plasmids, and cultural conditions**. All *E. coli* strains were grown at 37°C in Luria Bertani (LB) medium (solid or liquid). Plasmids were maintained in the *E. coli* strain EPI300 and *ccdB* survival strain DB3.1. *Xanthomonas oryzae* pv. *oryzae* strains were grown at 28°C in NA (Difco nutrient broth containing 3 g/L beef extract and 5 g/L peptone, 15 g/L agar for solidification, pH = 6.8). *Pseudomonas syringae* pv. *tomato* DC3000 and mutants were grown at 30°C in LB medium or King's B medium (peptone 20 g/L, $K_2HPO_4$ 1.5 g/L, $MgSO_4 \cdot 7H_2O$ 1.5 g/L, glycerol 10 ml/L, 15 g/l agar for solidification, pH = 7.2). *Erwinia amylovora* Ea9 and mutants were grown at 28°C in LB medium. *Agrobacterium tumefaciens* LBA4404 and mutant strains were grown on yeast extract and beef (YEB) medium (1 g/L yeast extract, 5 g/L peptone, 5 g/L sucrose, 5 g/L beef extract, 2 mM $MgSO_4$, and 15 g/L agar for solidification). Liquid and agar media were supplemented with spectinomycin (100 mg/L), tetracycline (10 mg/L), and kanamycin (50 mg/L) as needed. To evict the *sacB*-based base editor plasmid, 10% sucrose was used in the medium. Bacterial strains and plasmids are listed in Supplementary Table 1.

**DNA manipulation and plasmid construction**. pEN-dCas9-CDA1: gBlock fragments containing the recA and other gene promoter sequences and about 20 bp overlapping with the sequences of pEN-L4-PvirB-dCas9-UL-T3-R1 were synthesized by IDT (Integrated DNA Technologies, Coralville, USA). Detailed sequence information is provided in Supplementary Table 2. gBlock is cloned into pEN-L4-PvirB-dCas9-UL-T3-R1 at *Afl* II and *BstZ17* I through Gibson cloning. *attL4* gBlock was synthesized by IDT. *attL4* gBlock was inserted in individual new promoter-dCas9 deaminase plasmid at *Hpa* I site through Gibson cloning, resulting in pHrpXp-dCas9-CDA1-UGI, pRecAp dCas9-CDA1-UGI, pPIP2p-dCas9-CDA1-UGI, and pPIP3p-dCas9-CDA1-UGI. Detailed sequence information about gBlocks is listed in Supplementary Table 2.

The *attR3-attR4* cassette (1837 bp) from pPm43GW was cut out with *Hind* III and *Acc65* I and cloned into the predigested pHM1-Gib with the same enzymes. Correct clones were selected with spectinomycin and chloramphenicol antibiotics.

Target site selection: Premature stop codons could only be introduced by targeting the codons 5′-CAA-3′ (Gln), 5′-CAG-3′ (Gln), 5′- CGA -3′ (Arg) on the sense strand, or 5′-CCA-3′ (encoding Trp on the sense strand) on the antisense strand. The optimal window for C to T change is located at positions 16 to 20 bp upstream of PAM (5′-NGG-3′) and context effect adjacent to the target C is not so obvious. Therefore, the protospacer sequence needs to be adjusted depending on

the positions of targeted C to create a premature stop codon and PAM sequence. Accordingly, gRNAs were designed with CRISPR-CBEI[53] to introduce stop codons (CRISPR-STOP[54]) within the region of 16 to 20 bp upstream of the PAM sequence (5′-NGG-3′).

Complementary oligos (23 or 24 nucleotides) for gRNA with appropriate 4 nucleotide overhangs at the 5′ ends were synthesized. Oligonucleotides were annealed to form double stranded fragments (dsOligo) before cloning into the gRNA vector at the *Bsa* I site. All oligos corresponding to the spacer sequences of gRNA used in this study are listed in Supplementary Table 3. The dsOligos and the pEN-L1-PJ23119-BsaI-PglpT-sfGFP-TrrfB-SalI-Scaf-L2 vector were subjected to the Golden Gate assembly using *Bsa* I-HF and T4-DNA ligase. Then the reaction mix was electroporated into *E. coli*. Colonies carrying successful insertion of dsOligo were identified due to loss of green fluorescence, easily seen by the naked eye. The insertions were further confirmed by sequencing. The gRNA cassettes were combined with other entry vectors pEN-dCas9-CDA1, pEN-sacB and the destination vector pHM1attR3-attR4 (or pHM-GFP) through Gateway reaction according to the manufacturer's instructions. Correct plasmids were verified by restriction digestion using *Hind* III.

pTL-Bag vectors: Four gBlock fragments were synthesized by IDT. Each gBlock was cloned into pTL-N at *Xba* I and *Xho* I restriction sites through Gibson cloning, further confirm by sequencing, resulting to pTL-Bag1, pTL-Bag2, pTL-Bag3, and pTL-Bag4. To construct pTL-Bag2T, a fragment was PCR-amplified using oligos BagRNA2T-F and Seq-F and a template of the gBlock (gBlock-BagRNA4, about 220 bp for pTL-Bag4), then cloned into the pTL-N vector at *Xba* I and *Xho* I restriction sites through Gibson cloning, and further confirmed by sequencing.

The dsOligo fragments and individual gRNA shuttle vectors pTL-Bag1, pTL-Bag2, pTL Bag2T, pTL-Bag3, and pTL-Bag4 were subjected to Golden Gate assembly using *BsmB* I and T4-DNA ligase. The individual reactions were used to transform *E. coli*. Colony-PCR approach was used to screen for putative clones followed by sequencing using primer Seq-F, resulting in pTL-Bag1-gRNA1, pTL-Bag2-gRNA2, pTL-Bag2T-gRNA2, pTL-Bag3-gRNA3, pTL-Bag4-gRNA4. For two gRNA assembly, pTL-Bag1-gRNA1, pTL-Bag2T-gRNA2, and pEN-L1-PJ23119-PglpT-sfGFP-TrrfB-BsaI-Scaf-L2 vector were subjected to Golden Gate assembly using *Bsa* I-HF and T4-DNA ligase, followed by transformation of *E. coli*. Colony-PCR approach was used to screen for putative clones followed by sequencing using Seq-F, resulting in pEN-gRNA1 + gRNA2. For four gRNA assembly, pTL-Bag1-gRNA1, pTL-Bag2-gRNA2, pTL-Bag3-RNA3, pTL-Bag4-RNA4, and pEN-L1-PJ23119-BsaI-PglpT-sfGFP-TrrfB-SalI-Scaf-L2 vector were subjected to Golden Gate reaction using *Bsa* I-HF and T4-DNA ligase. The reaction was introduced into *E. coli*. Colony-PCR approach was used to select putative clones followed by sequencing to further confirm the accuracy using oligo Seq-F, resulting in pEN-gRNA1 + gRNA2 + gRNA3 + gRNA4.

Finally, the multiple gRNAs along with pEN-dCas9-CDA1-UGI, pEN-sacB, and pHMattR3-attR4 (or pHMattR3-attR4-GFP) were recombined into single plasmids through Gateway reaction. Resulting plasmids were verified by restriction digestion with *Hind* III.

To restore dead-Cas9 into Cas9 nickase (D10A) by converting alanine at position 840 back into histidine, overlapping PCR was conducted. pEN-dCas9-CDA1 was used as template to amplify the target region with appropriate primers (Supplementary Table 3). The amplicons were inserted back into pEN-dCas9-CDA1 at *Acc*65I and *BamH*I sites through Gibson cloning to result in pEN-nCas9-CDA1.

**Base editing, eviction of plasmid and genotyping of editing strains**. Base-editor plasmids were transferred into PXO99[A] (or DC3000, Ea9) through electroporation as described and modified from Choi et al.[55]. For selection, cells were plated on spectinomycin-containing NA for PXO99[A], King'B medium for DC3000 and LB medium for Ea9. Colony-PCR was first used to check for presence of base editing plasmids with single colonies before Sanger sequencing. For deep sequencing, five individual colonies were pooled together treated as one biological sample of three replicates for PCR-amplification of target regions. The target region was amplified with DNA polymerase and appropriate primers (Supplementary Table 3). Amplicons were treated with Exonuclease I and Shrimp Alkaline Phosphatase (New England Biolabs) for Sanger sequencing, or for deep sequencing using MiSeq Micro-PE150. The chromatograms were analyzed with SnapGene (GSL Biotech). To obtain the plasmid-free mutant strains, edited PXO99[A] (or DC3000, Ea9) strains were grown in 10% sucrose-containing liquid medium at 28°C for over 12 h, then plated on 10% sucrose medium. Single colonies were patched in duplicate on NA medium for PXO99[A] (King'B medium for DC3000 and LB medium for Ea9) and spectinomycin-containing NA medium (King'B medium for DC3000 and LB medium for Ea9). Bacteria having plasmid evicted was confirmed by the tolerance to sucrose and sensitivity to spectinomycin. The PCR-amplification of target region was sequenced again to confirm. The chromatograms were analyzed with Snap-Gene software (GSL Biotech). The editing efficiencies for gSuxB and gSuxC under four different promoters in *Xoo* were estimated by calculating the areas of peaks in chromatograms between C and edited T. The editing efficiencies for other gRNAs were assessed by amplicon deep sequencing.

Base-editor plasmids carrying gThyA were introduced to LBA4404 through electroporation modified from Choi et al.[55]. For selection, cells were plated on spectinomycin-containing LB medium with thymidine (50 mg/L). Single colonies were patched in duplicate on LB plates supplemented with thymidine and without

thymidine. The colonies could not grow on YEP without thymidine would be strains having plasmid evicted. The target region was PCR-amplified with appropriate primers (Supplementary Table 3). Amplicons were treated with Exonuclease I and Shrimp Alkaline Phosphatase (New England Biolabs) and then subjected to Sanger or deep sequencing. The chromatograms were analyzed with SnapGene software (GSL Biotech).

**Data analysis of deep sequencing reads**. Trim-galore (https://github.com/FelixKrueger/TrimGalore) was used to filter short reads by using a minimal quality Phred score ≥ 30. Additionally, adapters were clipped from the sequences. Mapping was conducting using Bowtie2(https://github.com/BenLangmead/bowtie2). Files of mapping results (sam files) were sorted and indexed with samtools (https://github.com/samtools/samtools) and variant detection was conducted with Pilon V1.2 (https://github.com/broadinstitute/pilon). Variant Call Files (VCFs) were parsed to obtain fields corresponding to: Position of the detected SNP, Reference base, SNP detected, Valid read depth, Counts of As, Cs, Gs, and Ts at the locus (field BC), and Allele Frequency (field AF).

**Pathogenicity assays**. *Arabidopsis* plants were grown in a growth chamber with a photoperiod of 8 h light/16 h dark at a temperature of 22 °C, under 75% humidity. For infiltration-based inoculation, the leaves of 4-week-old plants were infiltrated with a bacterial suspension of either $1 \times 10^6$ CFU ml$^{-1}$ (for visual disease assays) or $5 \times 10^4$ CFU ml$^{-1}$ (for in planta bacterial growth assays) in 10 mM MgCl2 using a needleless syringe. The surface of the leaves was blotted with a KimWipe to remove excess bacterial suspension. After the liquid inside the leaves was absorbed (about 1 h after infiltration), plants were returned to the growth chamber for the duration of the experiment. For both disease and bacterial growth assays, DC3000, $AvrPto^{cr-1}$, $AvrPto^{cr-2}$, $HopK^{cr-1}$ or $HopK^{cr-2}$ were each infiltrated into at least 24 leaves in 12 *Arabidopsis* plants per experiment. Disease assays were monitored for three days post inoculation (dpi), when photos were taken. For bacterial growth measurement, four-leaf disks (0.5 cm$^2$ in size) were collected from four different inoculated plants (one disk per plant) at 0 and 3 dpi. Leaf disks were ground in 10 mM MgCl2, and a dilution series was made prior to plating on *Pseudomonas* agar. Bacterial colonies were counted after two days incubation at 28 °C. The number for each time point represents the average of three measurements from each of three independent experiments (9 data points), which were statistically assessed at $p < 0.05$.

*N. benthamiana* plants were grown in a growth chamber maintained at 25 °C, relative humidity 60%, and light intensity ca. 80 umol/m/s. Leaves were perforated with a needle at the site of injection. *Erwinia* Ea9 and mutant suspensions were then infiltrated into the leaf using a needleless syringe. After infiltration, the cell death of leaves was monitored for about 24 h. For cell death (HR) experiments, fully expanded leaves were used until the plants had 12 to 14 leaves.

To evaluate the virulence of the *Xoo* WT and mutant strains, Kitaake rice plants were inoculated with bacteria at the age of 4–5 weeks using leaf-clipping method. The disease was scored by measuring the lesion length at 7 and 14 d after inoculation. Data were recorded from at least ten leaves, from which the mean and standard deviation were calculated.

**Whole-genome sequencing and SNV analysis**. PXO99[A] and edited strains, LBA4404 and its edited strains were sequenced using paired-end Illumina technology with a mean coverage of ~120X. Trim_galore software (https://www.bioinformatics.babraham.ac.uk/projects/trim_galore/) was used to process, and filter reads using Phred scores ≥ 30 as a cutoff. Bowtie2 aligner tool[56] was used to map reads against the sequence of PXO99[A] (accession number CP000967) and LBA4404. Freebayes (https://github.com/freebayes/freebayes) was used to call for SNPs and indels. Additionally, a Pilon polishing tool with a minimum of 30 X depth coverage was used to corroborate Freebayes calls. Reads mapping TAL effectors were not considered in the analysis. Additional manual curation and annotation of SNPs were conducted using Artemis (http://sanger-pathogens.github.io/Artemis/Artemis/) and SNP data software[57].

**Statistics and reproducibility**. Data are plotted by using the BoxPlotR (http://shiny.chemgrid.org/boxplotr). The boxplot is delimited by the first and the third quartile of the distribution of the studied variables. Whiskers extend 1.5 times the interquartile range from the 25th and 75th percentiles, outliers are represented by dots; crosses represent sample means; data points are plotted as blue circles. The total numbers (n) of sample points are indicated. Tukey's Honest Significant Difference test was employed for post-ANOVA pairwise tests for significance (set at $p < 0.05$), or alternatively, the two-sided Dunnett's test was used. The number of biological replicates is indicated in the legend of each figure.

**Reporting summary**. Further information on research design is available in the Nature Portfolio Reporting Summary linked to this article.

## Data availability

All study data are included in the article and/or SI Appendix. The uncropped agarose gel image for Supplementary Fig. 4a is provided as Supplementary Fig. 26. The whole

genome resequencing and deep sequencing data were deposited in NCBI under the Bioproject PRJNA915598. Plasmids were deposited in Addgene with following information: pHMattR3-attR4-GFP, 196247; pEN-dCas9-CDA1-UGI, 196248; pEN-sacB, 196249; pEN-L1-PJ23119-PplpT-sfGFP-TrrfB-BsaI-Scaf-L2, 196250; pTL-Bag1-gRNA1, 196251; pTL-Bag2-gRNA2, 196252; pTL-Bag2T-gRNA2, 196253; pTL-Bag3-gRNA3, 196254; pTL-Bag4-gRNA4, 196255.

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

## Acknowledgements
The work was partially supported by the USDA NIFA (2017-67013-26521 to B.Y.), subawards to University of Missouri and University of Florida from the Heinrich Heine University of Dusseldorf funded by the Bill & Melinda Gates Foundation (OPP1155704) (B.Y., F.F.W.), NSF (IOS-1741090 to F.F.W), and the China Scholar Council (C.L., as a joint PhD student).

## Author contributions
C.L. and B.Y. designed research; C.L., L.W., L.J.C., F.V., and J.H.T. performed research; W.G., L.P., H. D. and F.F.W. analyzed data; and C.L., F.F.W. and B.Y. wrote the paper with revision from other authors.

## Competing interests
The authors declare no competing interests.
