## [Peer Review File · Communications Biology]

Reviewers' comments:

Reviewer #1 (Remarks to the Author):

In this manuscript, the authors optimized the CBE systems in various bacterial pathogens. The demonstrated that the RecA promoter combined with pHM1 improved the base editing efficiency with up to 100% in four phytopathogenic bacterial species. Furthermore, the authors developed multiplex base editing vector and engineered CDA1-A deaminase with lower off-target effect. The above results may facilitate the functional analysis of the phytopathogenic bacteria. The authors have given a very good effort but still there are many issues which must be addressed before the acceptance of this manuscript.

Q1. Line 28, It should be "dCas9/nCas9", rather than Cas9.

Q2. The authors used Sanger sequence chromatograms to analyze the editing efficiency in a population, which is inaccurate. Amplicon deep sequencing should be performed to analyze the editing efficiency and mutation pattern in detail. In addition, the editing efficiencies were only exhibited in column chart, such as Fig. 2A, 2C and 2E. The sequence chromatograms should be provided in the Supporting Information.

Q3. In Fig.2F, multiple peaks were observed in the sequence chromatograms, which could not truly reflect the base editing outcomes. For example, if C-to-G occurred at C7 in the target sequence.

Q4. Multiplex base editing is interestingly in this work and more detailed information should be provided in the vector design for expression of the multiple sgRNAs. The sgRNAs were driven by several independent promoters or by single polycistronic cassette by a single promoter? If polycistronic cassette were selected, which RNA cleavage element was selected, Csy4 or tRNA?

Q5. In Fig.5B, the lesion lengths in rice leaves were exhibited by graphs. The authors should show the pictures of the lesion length caused by the respective Xoo strains.

Reviewer #2 (Remarks to the Author):

In this manuscript, Li et al. developed a versatile cytosine base editor (CBE) system which could achieve efficient C to T conversion in the genome of a series of phytopathogenic bacteria, including *Xanthomonas*, *Pseudomonas*, *Erwinia*, and *Agrobacterium*. In addition, the authors performed the multiplexing base editing and demonstrated the high editing efficiency in *Pseudomonas* and *Xanthomonas*. Finally, they assessed the off-target effect of CBE in the editing bacteria, and tried to improve the fidelity by engineering of CDA1 variants.

Although the base editors have been widely used in various organisms, its application in plant pathogenic bacteria is still limited. In this study, the authors provided a simple and efficient genome editing tool; it would accelerate a wide variety of investigations in phytopathogenic bacteria. Here are some questions and modifications that should be addressed before acceptance.

1. As shown in the manuscript, some Cs in the protospacer were successfully edited, while some Cs were not. Could the authors summarize the editing rule of this base editor (What's the editable window? Does the adjacent base affect the editing efficiency?). It would be valuable for readers when designing the guide RNA.

2. In this study, the author used the dCas9 in CBE. Could the authors try nCas9 (Cas9D10A) which was reported to improve the editing efficiency (Komor et al. *Nature*, 2016, 533: 420)?

3. To assess the editing activity of CDA1 variants, the *thyA* gene was selected as the target site. The editing efficiency was evaluated according to the survival rate of *Agrobacterium*. To confirm that CDA1-A maintains the similar on-target editing activity as the wild type CDA1, it is better to test more editing sites.

4. Line 68-69, the authors say "Use of base editors, specifically, in phytopathogenic bacteria has been limited to *Agrobacterium*". To my knowledge, the cytosine base editor has been applied to *Pseudomonas syringae* (Chen et al. *iScience*, 2018, 6:222).

5. Line 388-390, the authors need to describe more detailedly how to measure the editing efficiency.

6. Figure 4A. The red arrow represents the target region of gRNA. Therefore, the red arrow of gRNA1 should locate directly below Q239 and Q240. In the figure legend, the sentence "The numbers indicate the positions of triplets in the protospacer sequences (in red)" is puzzled. Does it mean "The numbers indicate the positions of mutated residues in the dspA/E gene ORF?"
7. Figure S3A. What does the "Ho" and "He" mean? The authors need to describe them in the figure legend.
8. Figure S15. It seems that the authors performed a serial dilution. It is better to label the dilution fold in the figure.
9. Line 375. "Pseudomonas syringe" should read "Pseudomonas syringae".

Responses point-by-point

First, I would like to thank the two expert reviewers for evaluating our manuscript and providing the constructive comments. We have addressed all the concerns experimentally and sincerely hope that the revision and responses are satisfactory for acceptance of our revised manuscript for publication in *Communications Biology*. Please see point-by-point responses (in blue) below.

Reviewers' comments:

Reviewer #1 (Remarks to the Author):

In this manuscript, the authors optimized the CBE systems in various bacterial pathogens. The demonstrated that the RecA promoter combined with pHM1 improved the base editing efficiency with up to 100% in four phytopathogenic bacterial species. Furthermore, the authors developed multiplex base editing vector and engineered CDA1-A deaminase with lower off-target effect. The above results may facilitate the functional analysis of the phytopathogenic bacteria. The authors have given a very good effort but still there are many issues which must be addressed before the acceptance of this manuscript.

Q1. Line 28, It should be “dCas9/nCas9”, rather than Cas9.

Response

Changed into “A gene for a dCas9 or nCas9,..”.

Q2. The authors used Sanger sequence chromatograms to analyze the editing efficiency in a population, which is inaccurate. Amplicon deep sequencing should be performed to analyze the editing efficiency and mutation pattern in detail. In addition, the editing efficiencies were only exhibited in column chart, such as Fig. 2A, 2C and 2E. The sequence chromatograms should be provided in the Supporting Information.

Response

Additional amplicon deep sequencing experiments were performed to determine the editing efficiency of 10 individual gRNAs along with the *recA* promoter driven dCas9 and 8 individual gRNAs along with *recA* promoter driven nCas9 in *Xanthomonas*, *Pseudomonas* and *Erwinia*. Five transformants (in triplicates) for each target site were pooled together at day 4 before evicting the plasmids to extract genomic DNA for PCR amplification. The amplicons (54 samples in total) were subjected to MiSeq Micro-PE150 sequencing. The new data are presented in the revised manuscript in text figures and supplementary figures. The two approaches revealed the similar results at the optimal editing sites.

The chromatograms for gSuxC and gSuxB are provided in Suppl. Fig. S2. The editing efficiency for Fig. 2C and 2E are replaced by the amplicon deep sequencing data still as Fig. 2C and 2E.

Q3. In Fig.2F, multiple peaks were observed in the sequence chromatograms, which could not truly reflect the base editing outcomes. For example, if C-to-G occurred at C7 in the target sequence.

Response

The sequencing chromatogram for Fig. 2F contained some noise (e.g., C-to-G change) of sequencing peak as such noise occurred in the whole chromatogram of that sample. We changed another chromatogram in Fig. 2F.

Q4. Multiplex base editing is interestingly in this work and more detailed information should be provided in the vector design for expression of the multiple sgRNAs. The sgRNAs were driven by several independent promoters or by single polycistronic cassette by a single promoter? If polycistronic cassette were selected, which RNA cleavage element was selected, Csy4 or tRNA?

Response

The modular multiple gRNAs were described in more detail in the revised manuscript (Fig. S12 and Table S2). Basically the same promoter J23119 is used for each gRNA expression.

Q5. In Fig.5B, the lesion lengths in rice leaves were exhibited by graphs. The authors should show the pictures of the lesion length caused by the respective Xoo strains.

Response

The pictures of leaves showing the lesions caused by the individual *Xoo* strains were provide in Fig. 5B.

Reviewer #2 (Remarks to the Author):

In this manuscript, Li et al. developed a versatile cytosine base editor (CBE) system which could achieve efficient C to T conversion in the genome of a series of phytopathogenic bacteria, including *Xanthomonas*, *Pseudomonas*, *Erwinia*, and *Agrobacterium*. In addition, the authors performed the multiplexing base editing and demonstrated the high editing efficiency in *Pseudomonas* and *Xanthomonas*. Finally, they assessed the off-target effect of CBE in the editing bacteria, and tried to improve the fidelity by engineering of CDA1 variants. Although the base editors have been widely used in various organisms, its application in plant pathogenic bacteria is still limited. In this study, the authors provided a simple and efficient genome editing tool; it would accelerate a wide variety of investigations in phytopathogenic bacteria.

Here are some questions and modifications that should be addressed before acceptance.

1. As shown in the manuscript, some Cs in the protospacer were successfully edited, while some Cs were not. Could the authors summarize the editing rule of this base editor (What's the editable window? Does the adjacent base affect the editing efficiency?). It would be valuable for readers when designing the guide RNA.

Response

A section of guide RNA target site has been added in the Method section in the revised manuscript.

2. In this study, the author used the dCas9 in CBE. Could the authors try nCas9 (Cas9D10A) which was reported to improve the editing efficiency (Komor et al. Nature, 2016, 533: 420)?

Response

nCas9 has been constructed and used for 9 gRNAs. The editing efficiencies as estimated by amplicon deep sequencing is similar to those mediated by dCas9 (Fig. S10 and S11).

3. To assess the editing activity of CDA1 variants, the *thyA* gene was selected as the target site. The editing efficiency was evaluated according to the survival rate of *Agrobacterium*. To

confirm that CDA1-A maintains the similar on-target editing activity as the wild type CDA1, it is better to test more editing sites.

Response

We have performed additional experiments to obtain more clones (15 transformants) of *thyA* targeted by CDA1 and CDA-1, and used amplicon deep sequencing to reassess the efficiency, which showed the similar frequency as previously – Fig. S23. We also provided data from another experiment of multiplex base editing of four *xop* genes using CDA1-A in *Xanthomonas campestris* pv. *campestris* (Fig. S24).

4. Line 68-69, the authors say “Use of base editors, specifically, in phytopathogenic bacteria has been limited to *Agrobacterium*”. To my knowledge, the cytosine base editor has been applied to *Pseudomonas syringae* (Chen et al. *iScience*, 2018, 6:222).

Response

The paper by Chen et al. has been cited and added.

5. Line 388-390, the authors need to describe more detailedly how to measure the editing efficiency.

Response

More detail has been added in the revised manuscript. “The editing efficiencies for gSuxB and gSuxC under four different promoters in *Xoo* were estimated by calculating the areas of peaks in chromatograms between C and edited T. The editing efficiencies for other gRNAs were assessed by amplicon deep sequencing.”

6. Figure 4A. The red arrow represents the target region of gRNA. Therefore, the red arrow of gRNA1 should locate directly below Q239 and Q240. In the figure legend, the sentence “The numbers indicate the positions of triplets in the protospacer sequences (in red)” is puzzled. Does it mean “The numbers indicate the positions of mutated residues in the *dspA/E* gene ORF?”

Response

The red arrow for gRNA1 has been shifted directly beneath the Q239 and 240. The legend has also been corrected – “The numbers indicate the positions of triplets in the *dspA/E* coding sequence (in red).”

7. Figure S3A. What does the “Ho” and “He” mean? The authors need to describe them in the figure legend.

Response

The description of “Ho” for homogenous edits and “He” for heterogenous edits is provided in the figure legend.

8. Figure S15. It seems that the authors performed a serial dilution. It is better to label the dilution fold in the figure.

Response

The 10X serial dilution is added in the figure legend.

9. Line 375. “*Pseudomonas syringe*” should read “*Pseudomonas syringae*”.

Response

Corrected.

REVIEWERS' COMMENTS:

Reviewer #1 (Remarks to the Author):

The authors addressed all my concerns. The current manuscript is ready for publication.

Reviewer #2 (Remarks to the Author):

The revised manuscript is improved across the board with edits, revisions, additions, and clarifications to the text and figures. In general, the authors have addressed all my concerns.